

# Salt and osmotic stress can improve the editing efficiency of CRISPR/Cas9-mediated genome editing system in potato

Mingwang Ye[1,2,3], Mengfan Yao[1,2,3], Canhui Li[3,4] and Ming Gong[1,2,3,5]

[1] School of Life Sciences, Yunnan Normal University, Kunming, Yunnan, China
[2] Engineering Research Center of Sustainable Development and Utilization of Biomass Energy, Ministry of Education, Yunnan Normal University, Kunming, Yunnan, China
[3] Yunnan Key Laboratory of Potato Biology, Yunnan Normal University, Kunming, Yunnan, China
[4] Joint Academy of Potato Science, Yunnan Normal University, Kunming, Yunnan, China
[5] Key Laboratory of Biomass Energy and Environmental Biotechnology of Yunnan Province, Yunnan Normal University, Kunming, Yunnan, China

Corresponding author
Ming Gong,
gongming6307@163.com

## ABSTRACT

CRISPR/Cas9-mediated genome editing technology has been widely used for the study of gene function in crops, but the differences between species have led to widely varying genome editing efficiencies. The present study utilized a potato hairy root genetic transformation system and incorporated a rapid assay with GFP as a screening marker. The results clearly demonstrated that salt and osmotic stress induced by NaCl (10 to 50 mM) and mannitol (50 to 200 mM) treatments significantly increased the positive rates of genetic transformation mediated by *A. rhizogenes* and the editing efficiency of the CRISPR/Cas9-mediated genome editing system in potato. However, it was observed that the regeneration of potato roots was partially inhibited as a result. The analysis of CRISPR/Cas9-mediated mutation types revealed that chimeras accounted for the largest proportion, ranging from 62.50% to 100%. Moreover, the application of salt and osmotic stress resulted in an increased probability of null mutations in potato. Notably, the highest rate of null mutations, reaching 37.5%, was observed at a NaCl concentration of 10 mM. Three potential off-target sites were sequenced and no off-targeting was found.

In conclusion, the application of appropriate salt and osmotic stress significantly improved the editing efficiency of the CRISPR/Cas9-mediated genome editing system in potato, with no observed off-target effects. However, there was a trade-off as the regeneration of potato roots was partially inhibited. Overall, these findings present a new and convenient approach to enhance the genome editing efficiency of the CRISPR/Cas9-mediated gene editing system in potato.

## INTRODUCTION

CRISPR/Cas9 (clustered regularly interspaced short palindromic repeats/CRISPR-associated protein 9) is a novel genome editing technology based on the natural immune system of bacteria: Cas9 nuclease, single guide RNA (sgRNA) and target DNA sequences.

By using Cas9 nuclease and sgRNA in combination to target specific DNA sequences, precise cleavage and modification of the target sequence is achieved (*Jinek et al., 2012*). Because of many potential targets and the convenience of vector construction, CRISPR/ Cas9 has been widely used to verify gene function in crops (*Shan et al., 2013*; *Butler et al., 2015*; *Ren et al., 2019*). The efficiency of the CRISPR/Cas9-mediated genome editing system; however, varies widely among different species (*Kusano et al., 2018*; *Ye et al., 2018*).

Currently, the studies on improving genome editing efficiency are mainly focused on the CRISPR/Cas9 system itself. For Cas9, the codon and promoter optimization of the Cas9 gene expressed in specific crops can be used to improve the expression levels of Cas9, thus improving genome editing efficiency (*Wang et al., 2015*; *Eid, Ali & Mahfouz, 2016*; *Feng et al., 2018a*; *Feng et al., 2018b*). In the case of sgRNA, gene editing efficiency can be improved by optimizing the design of gRNA (*Liang et al., 2016*) or by designing multiple target sites for a single gene (*Cong et al., 2013*; *Brooks et al., 2014*; *Hashimoto et al., 2018*). In addition, some studies have shown that editing efficiency can also be improved by heat stress (*LeBlanc et al., 2018*; *Nandy et al., 2019*). Beyond that, little attention has been paid to the effect of other abiotic stresses on genome editing efficiency.

Abiotic stresses such as, osmotic, salt, drought, heat and cold can affect plant growth, development and reproduction, and long-term abiotic stress can lead to plant death (*Lima et al., 2015*; *Basu et al., 2016*). During abiotic stress, excessive accumulation of reactive oxygen species leads to oxidative stress. Although reactive oxygen species are important signaling molecules involved in regulating plant growth, development and stress response, due to its high reactivity, excessive accumulation of reactive oxygen species will inevitably attack biological macromolecules, including DNA (*Raja et al., 2017*), thus it can lead to the instability of the genomic DNA (*Chiera et al., 2008*). Therefore, we wondered if abiotic stresses would affect genome editing efficiency.

Potato is an important crop due to its nutritional value and high yield potential, and is also considered a global food security crop. Genome editing will provide efficient and rapid technical approaches for targeted trait improvement and precise molecular breeding in potato (*Ye, Li & Ming, 2020*; *Ahmad et al., 2022*; *Tuncel & Qi, 2022*). Our previous work showed that knocking out the self-incompatibility gene S-RNase by using the CRISPR/ Cas9 system could create self-compatible diploid potato, which opens new avenues for diploid potato breeding and will also be useful for studying other self-incompatible crops (*Ye et al., 2018*). However, among different potato germplasms and with different CRISPR/Cas9 systems, the editing efficiency can vary greatly. *Kusano et al. (2018)* demonstrated that by utilizing the translational enhancer dMac3 and multiple guide RNAs in a modified CRISPR/Cas9 system, the editing efficiency could reach as high as 75% when using the tetraploid *Solanum tuberosum* L. cultivar Sayaka as experimental material (*Kusano et al., 2018*). On the other hand, our experiments using the diploid germplasms *S. tuberosum* group *Phureja* yielded an editing efficiency of only 17.7% (*Ye et al., 2018*). Although loss of function mutagenesis through conventional SpCas9 targeting has proven to be essential for functional genomics and for altering traits to a certain extent, more

PeerJ ___________________________________________________

efficient, precise, and diverse CRISPR/Cas systems are needed to be tested in potato (*Tiwari et al., 2022*; *Tuncel & Qi, 2022*).

*Agrobacterium rhizogenes* (*A. rhizogenes*) is a Gram-negative soil bacterium that can infect almost all dicotyledonous plants and a few monocotyledonous plants. It carries a T-DNA fragment from the Ri plasmid into the plant cell and integrates into the plant genome, stimulating the plant to produce a large number of hairy roots (*Bahramnejad et al., 2019*). Compared to *Agrobacterium tumefaciens*, *A. rhizogenes* can induce transgenic roots more rapidly (*Liu et al., 2016*). Therefore, *A. rhizogenes*-mediated genetic transformation in plant species becomes a simple, rapid and efficient way to learn gene functions (*Veena & Taylor, 2007*).

In the present study, we demonstrated a modified method in which the genome editing efficiency of transgenic potato plants can be enhanced under proper sodium chloride (NaCl)–mediated salt stress and mannitol–mediated osmotic stress with no off-targeting effect, which provides a new and convenient approach for studying the genome editing efficiency of the CRISPR/Cas9-mediated gene editing system in potato.

# MATERIALS AND METHODS

## Plant materials

The diploid potato germplasm CIP 149 (*S. tuberosum* group *Phureja*) used in this study was kindly provided by International Potato Center (CIP). Potato virus-free seedlings were grown cultured on solid Murashige and Skoog (MS) medium (pH 5.8) with 3% (w/v) sucrose under the conditions of long-day (16 h light/8 h dark) at $25 \pm 1$ °C.

## Plasmid constructs

The binary vector pKESE401 expressing SpCas9 and intermediate vector pCBC-DT1T2 were used in this experiment as described previously (*Ye et al., 2018*). The report gene CaMV35S::eGFP inserted in pKESE401 which was digested by EcoRI to generate plasmid pKESE402. The gene sequences of *StFMO GS-OX-Like3* (*StLike3*) of potato (gene ID: PGSC0003DMT400035866) were downloaded from the spud DB database (http://solanaceae.plantbiology.msu.edu). The target gRNA sequence of *StLike3* was selected using the CRISPR-P tool 2.0 (http://crispr.hzau.edu.cn/CRISPR2/), then incorporated into the pKSE402 backbone using the protocol (*Xing et al., 2014*). The primers used to insert the sgRNA into the pKSE402 vector are shown in Table 1.

## *A. rhizogenes* transformation of potato

*A. rhizogenes* transformation followed the protocol reported previously (*Butler, Jansky & Jiang, 2020*), with some modifications as follows: potato internodal segments approximately 1 cm excised from 4-weeks plantlets were used for transformation. After 2 days of pre-cultivated on MS (Phytotechlab, Lenexa, KS, USA) solid medium containing 2 mg/L α-naphthaleneacetic acid (Sigma-Aldrich, St. Louis, MO, USA) and 1 mg/L zeatin (Phytotechlab, Lenexa, KS, USA), these explants were infected with *A. rhizogenes* (Ar.Qual, Frdbio, Wuhan, China) harbouring pKSE402 with the target sequence (OD600≈0.5) for 10 min, then incubated on the solid medium in the presence of 2 mg/L α-naphthaleneacetic

**Table 1 Primers used in the experiments.**

| Primers | Sequence (5'–3') | Application |
|---|---|---|
| L3-402BSA-F | ATTGTTTGTGGCCAAGAAAAAGCC | Construction of *Like3* sgRNA |
| L3-402BSA-R | AAACGGCTTTTTCTTGGCCACAAA | |
| U626-IDF | TGTCCCAGGATTAGAATGATTAGG | Detection of the exogenous genes |
| U626-IDR | AACCCCAGAAATTGAACGCC | |
| L3-F | TCCCATTTTCCCATCCTCTCT | Detection of the transgenic roots |
| L3-R | TCTTTCCCACAAACCCCACT | |
| OT-1-F | CGCATCCCTCCTTAACTTGC | Detection of Like3 off-target site 1 |
| OT-1-R | TGCATAGGGAAGACGAAGCT | |
| OT-2-F | CCAGAGCTTTCTTGGCCTTG | Detection of Like3 off-target site 2 |
| OT-2-R | AGTCTAGAACACTTGGGAGCC | |
| OT-3-F | ACCCGAGTTTTCCCCTTTTG | Detection of Like3 off-target site 3 |
| OT-3-R | TGAGTTAATGGTGGAAAGCCC | |

acid and 1 mg/L zeatin for 2 days under dark condition. Finally, the explant segments were transferred to MS solid medium containing 200 mg/L Timentin, and 0, 10, 20, 30 and 50 mM of NaCl or 0, 50, 100, 150 and 200 mM of Mannitol, respectively. The roots were proliferated after 15–20 days and used for following analysis.

## Transformation effciency detection

To confirm the transformants of roots, stereo fluorescence microscope was used to observe their GFP expression. Genomic DNA were extracted using the standard cetyltrimethylammonium bromide (CTAB) method. The presence of T-DNA in positive roots were confirmed by PCR using specific primers (Table 1). The positive rate was calculated by determining the proportion of transformants present in the regenerated roots. Additionally, the transformation efficiency was calculated based on the electrophoresis amplicons of PCR products.

## Mutation effciency detection

The target sites of transgenic roots were amplified by specific primers (Table 1). PCR amplicons were cloned into pBM16A vector (Biomed, Beijing, China), then transformed *E. coli* competent cells. About 5–7 colonies were selected from each plate that contained the DNA of positive transgenic roots for sanger sequencing (Sangon Biotech, Shanghai, China). Geneious 4.8.3 software was used to analyze the sequences, the mutation rates were calculated, and the mutation type and genotype were analyzed based on all sequence data.

## Off-target analysis

Potential off-target sites (OTs) were automatically generated using the CRISPR-P tool 2.0 when selecting the target site. For the *StLike3* sgRNA, the top three OTs were selected for analysis. The specific primers targeting the potential OTs are shown in Table 1. Five transgenic roots were randomly selected from each treatment: control, 20 mM NaCl and

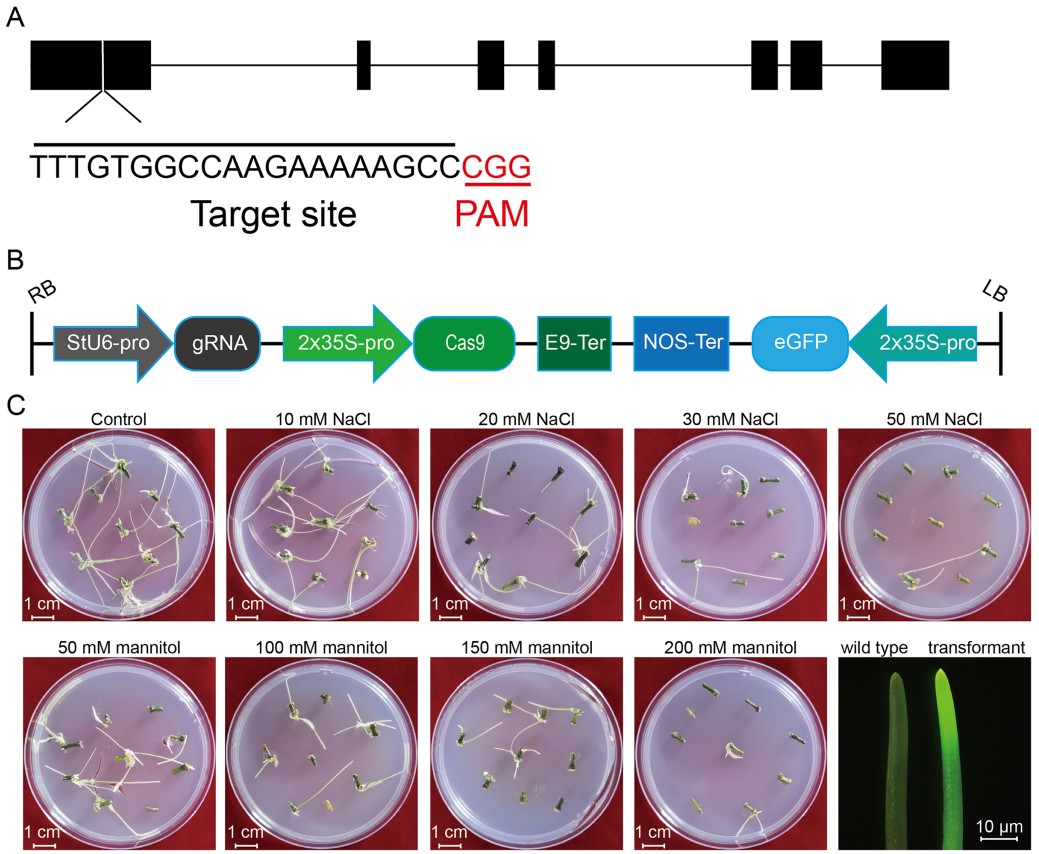

**Figure 1 Schematic diagram of the CRISPR/cas9 vector and the proliferation of potato roots under salt and osmotic stress using A. *rhizogenes* transformation.** (A) The schematic diagram of the gene *StLike3* with the target site, the gRNA sequence was under-lined by black line, and PAM was marked by red line. (B) Structure of the CRISPR/Cas9 binary vector for potato transformation. (C) Observation of the root proliferation under salt and osmotic stress.

100 mM mannitol, respectively, and about 5–7 colonies per transgenic root were randomly picked out for sequencing and further analysis.

## RESULTS

### Effect of salt and osmotic stress on genetic transformation and proliferation of potato roots

To investigate the effect of salt or osmotic stress on genetic transformation efficiency mediated by the CRISPR/Cas9 system, a gRNA target site on the first exon of *StLike3*, which is involved in catalyzing specific steps in the metabolism of glucosinolates (*Schlaich, 2007*), was selected (Fig. 1A), CRISPR/cas9 vector pKSE402 was constructed as shown in Fig. 1B, and the *A. rhizogenes* harboring pKSE402 was used to co-incubate with potato explants to obtain transformants. In the present experiments, 54 explants were co-incubated in each treatment. After 20 d of cultivation, the proliferating roots of the explants were observed under a stereo-fluorescence microscope and the roots expressing GFP were selected (Fig. 1C). Their DNA was extracted for T-DNA detection and the

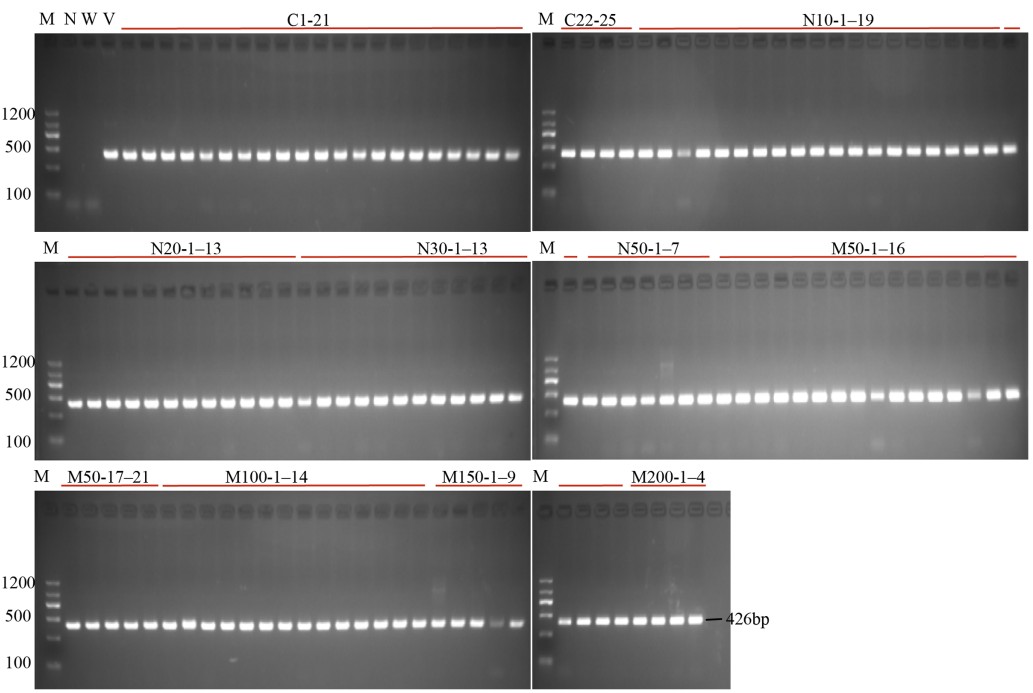

**Figure 2 PCR detection of the U6 promoter for the investigation of transformation efficiency.** M, marker; N, negative control; W, wild-type; V, positive control; C1–25, the regeneration roots with no osmotic stress; N10-1–19, N20-1–13, N30-1–13, N50-1–7: indicated the roots number which were regenerating on the medium with 10, 20, 30 and 50 mM NaCl; M50-1–21, M100-1–14, N150-1–9, N200-1–4: indicated the roots number which were regenerating on the medium with 50, 100, 150 and 200 mM mannitol.

**Table 2 Effect of salt and osmotic stress on root proliferation and genetic transformation efficiency in potato.**

| Treatments (mM) | | No. of explants | No. of roots | Root induction efficiency(%) | No. of transformant roots | Transformant efficiency(%) | Positive rates (%) |
|---|---|---|---|---|---|---|---|
| Control | 0 | 54 | 163 | 301.85 | 25 | 46.30 | 15.34 |
| NaCl | 10 | 54 | 122 | 225.93 | 19 | 35.19 | 15.57 |
| | 20 | 54 | 81 | 150.00 | 13 | 24.07 | 16.05 |
| | 30 | 54 | 25 | 46.30 | 13 | 24.07 | 52.00 |
| | 50 | 54 | 11 | 20.37 | 7 | 12.96 | 63.64 |
| Mannitol | 50 | 54 | 133 | 246.30 | 21 | 38.89 | 15.79 |
| | 100 | 54 | 81 | 150.00 | 14 | 25.93 | 17.28 |
| | 150 | 54 | 35 | 64.81 | 9 | 16.67 | 25.71 |
| | 200 | 54 | 10 | 18.52 | 4 | 7.41 | 40.00 |

results showed that all transformants contained the T-DNA (Fig. 2). The number of regenerated roots were counted and genetic transformation efficiency was calculated (Table 2). The results showed that both the regeneration and transformation efficiency of potato roots decreased with increasing concentrations of NaCl and mannitol. However, the positive rates of the roots clearly increased as the concentrations of NaCl and mannitol

**Table 3 Effect of salt and osmotic stress on genome editing efficiency and mutation types in potato roots.**

| Treatments (mM) | No. of roots examined | No. of roots with mutations | Mutation rate (%) | Homozygous | | Bi-allelic | | Heterozygous | | Chimeric | |
|---|---|---|---|---|---|---|---|---|---|---|---|
| | | | | Number | Ratio (%) | Number | Ratio (%) | Number | Ratio (%) | Number | Ratio (%) |
| Control | 25 | 18 | 72.00 | 0 | 0.00 | 2 | 11.11 | 0 | 0.00 | 16 | 88.89 |
| N10 | 19 | 16 | 84.21 | 1 | 6.25 | 5 | 31.25 | 1 | 6.25 | 9 | 56.25 |
| N20 | 13 | 13 | 100.00 | 0 | 0.00 | 4 | 30.77 | 2 | 15.38 | 7 | 53.85 |
| N30 | 13 | 13 | 100.00 | 0 | 0.00 | 0 | 0.00 | 0 | 0.00 | 13 | 100.00 |
| N50 | 7 | 7 | 100.00 | 0 | 0.00 | 0 | 0.00 | 0 | 0.00 | 7 | 100.00 |
| M50 | 21 | 18 | 85.21 | 0 | 0.00 | 1 | 5.56 | 2 | 11.11 | 15 | 83.33 |
| M100 | 14 | 14 | 100.00 | 0 | 0.00 | 2 | 14.29 | 2 | 14.29 | 10 | 71.43 |
| M150 | 9 | 9 | 100.00 | 0 | 0.00 | 1 | 11.11 | 0 | 0.00 | 8 | 88.89 |
| M200 | 4 | 4 | 100.00 | 0 | 0.00 | 0 | 0.00 | 0 | 0.00 | 4 | 100.00 |

**Note:**
N#, the concentration of NaCl (mM); M#, the concentration of mannitol (mM).

increased, reaching their highest at a concentration of 50 mM NaCl with a positive rate of 63.64%, and at a mannitol concentration of 200 mM with a rate of 40.00%, respectively (Table 2). The above-mentioned results clearly indicated that the salt and osmotic stress could greatly enhance the positive rates mediated by *A. rhizogenes* although the regeneration of potato roots and genetic transformation efficiency was inhibited to some extent.

## Analysis of mutation efficiency and types

Since the salt and osmotic stress could greatly improve the positive rates of potato roots (Table 2), the genome editing efficiency and mutation types of CRISPR/Cas9-mediated *StLike3* under these stresses were further investigated (Tables S1 and S2). The results showed that the genome editing efficiency of *StLike3* increased with increasing NaCl and mannitol concentrations, and all regenerated positive roots contained various mutations when NaCl concentration was equal to or more than 20 mM and mannitol was 100 mM (Table 3). In addition, all the sequencing results were counted as shown in Fig. 3. when NaCl concentration was more than 20 mM and mannitol was more than 100 mM, the mutation efficiency was greater than 75%. Among them, the mutation efficiency reached the maximum when the NaCl concentration was 50 mM, which was 91.67% (Fig. 3). Among all of these mutations, chimeras accounted for the largest proportion of all mutation types, but bi-allelic and homozygous mutations could be detected. In fact, the percentage of bi-allelic mutation was 37.5% and 21.43% at 10 mM NaCl or 100 mM mannitol, respectively, and homozygous mutations were also found under 10 and 30 mM NaCl stress (Tables 3, S1, and S2). The probability of null mutations (homozygous + bi-allelic + partial chimeric) was then statistically analyzed (Fig. 4). It can be seen that the probability of null mutations is higher for all treatment types than for the control. Specifically, the probability of null mutations reached a maximum at 50 mM NaCl concentration and 200 mM mannitol concentration, with values of 71.43% and 55.56%,

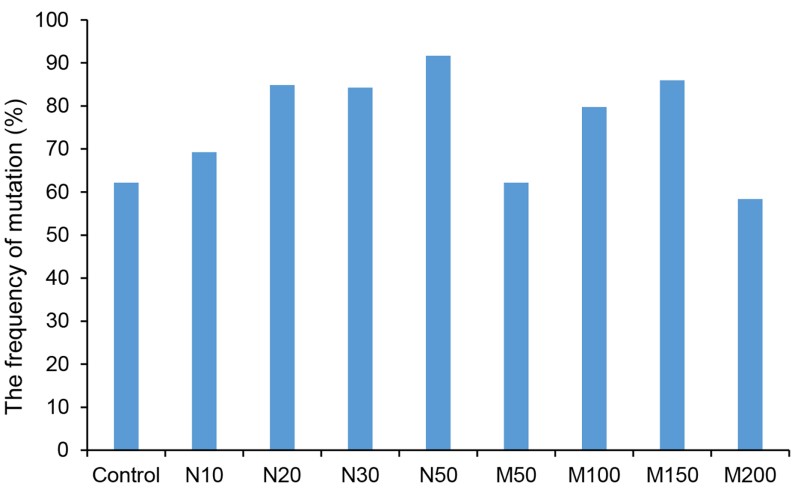

**Figure 3** The ratio of CRISPR/Cas9 system-induced *StLike3* mutation in potato roots under salt and osmotic stress. N#, the concentration of NaCl (mM); M#, the concentration of mannitol (mM).

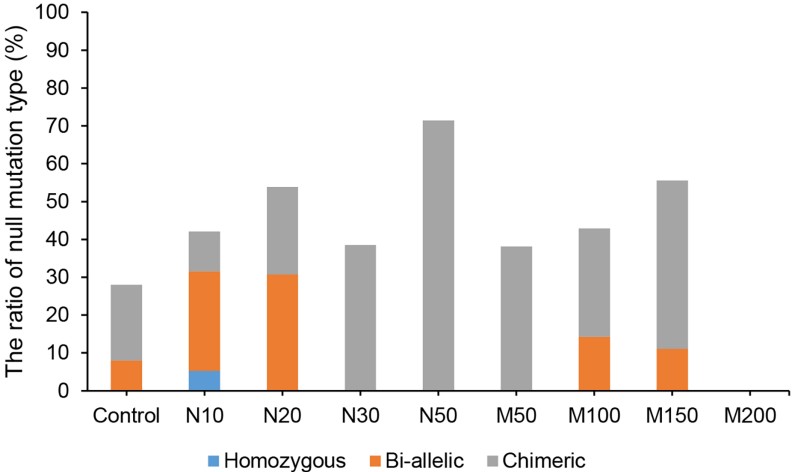

**Figure 4** The proportion of CRISPR/Cas9-mediated *StLike3* null mutations in potato roots under salt and osmotic stress. N#, the concentration of NaCl (mM); M#, the concentration of mannitol (mM).

respectively. It implied that these stresses were able to increase the null mutation probability.

## Patterns and frequency of CRISPR/Cas9-mediated *StLike3* mutations

Among all of the CRISPR/Cas9-mediated *StLike3* mutation types under salt and osmotic stress, deletions accounted for the largest proportion. The rate of deletion mutations reached its peak, at 92.86%, at a mannitol concentration of 200 mM (Fig. 5 and Table S1). The longest deletion, amounting to 213 base pairs, occurred at a mannitol concentration of 150 mM (Table S1, M150-1). All insertion mutations were only 1–2 bp, and a replacement

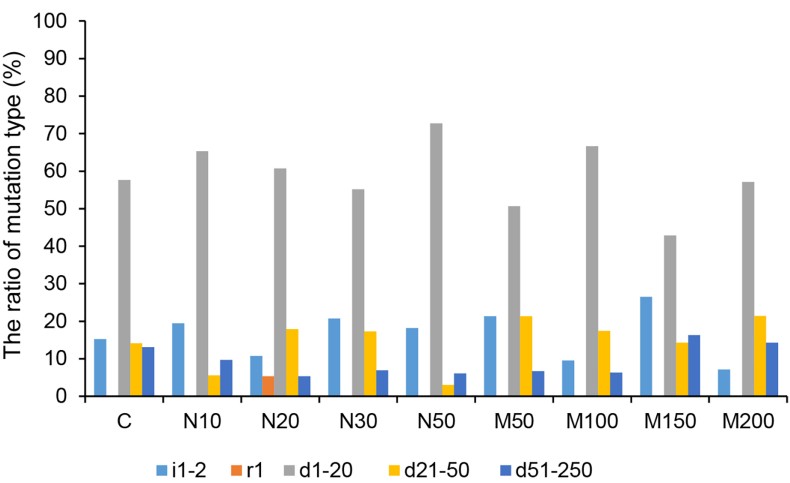

**Figure 5 The ratio of CRISPR/Cas9 system-induced *StLike3* mutation types in potato roots under salt and osmotic stress.** N#, the concentration of NaCl (mM); M#, the concentration of mannitol (mM). All sequences are followed by a description for the mutation type: i, insertion; r, replacement; d, deletion. The number associated with i, r, and d indicates the number of base pair changes.

**Table 4 Effect of salt and osmotic stress on the off-targeting probability of the selected three off-target sites of *StLike3* sgRNA.**

| Off-target sites | Sequence* | Mis-matches | Salt or osmotic stress (mM) | | No. of cloning examined | No. of off-targeting |
|---|---|---|---|---|---|---|
| OT-1 | TTTGTGGC**A**AAGAAAAA**ATT**AGG | 4 | Control | 0 | 31 | None |
| | | | NaCl | 20 | 34 | None |
| | | | Mannitol | 100 | 33 | None |
| OT-2 | **GTG**GTGGCCAAGA**G**AAA**A**CCTGG | 4 | Control | 0 | 30 | None |
| | | | NaCl | 20 | 27 | None |
| | | | Mannitol | 100 | 32 | None |
| OT-3 | TTT**TAGA**CCAAGAAAAAGC**A**AGG | 4 | Control | 0 | 29 | None |
| | | | NaCl | 20 | 28 | None |
| | | | Mannitol | 100 | 27 | None |

**Note:**
*Red fonts are mismatches. PAM is underlined.

mutation was only found at a NaCl concentration of 20 mM, which involved the replacement of one base pair (Fig. 5 and Table S1, N20-13).

## Detection of the rate of off-targeting in transgenic roots

Since the salt and osmotic stress could improve the CRISPR/Cas9-mediated genome editing efficiency (Table 3), we further investigated whether these abiotic stresses could enhance the off-targeting probability of the CRISPR/Cas9-mediated genome editing. The potential top three OTs were selected based on CRISPR-P tool 2.0 (Table 4), and five transgenic roots were randomly picked out from each treatment: control, 20 mM NaCl and 100 mM mannitol, respectively. 5–7 colonies per transgenic root were randomly picked

out for sequencing, and the sequencing data were statistically analyzed (Table 4). The results showed that no mutations in the three OTs could be found (Table 4).

## DISCUSSION

Since 2012, when CRISPR/Cas9-mediated genome editing technology was first applied to genome editing of living cells (*Jinek et al., 2012*), scientists have optimized and improved the technology in many ways for genome editing efficiency. In addition to optimizing the CRISPR/Cas9 system itself, the environment in which the organism resides can also be optimized to increase genome editing efficiency (*LeBlanc et al., 2018*). In *Arabidopsis* and *Citrus*, CRISPR/Cas9-mediated genome editing was able to increase the editing efficiency of somatic and germ cells under 37 °C stress by 5-fold and 100-fold, respectively, compared to standard temperature (22 °C) (*LeBlanc et al., 2018*).

In the present study, we used a potato hairy root genetic transformation system and added GFP as a screening marker for rapid assay. Because of their good solubility and ease of addition to the culture medium, NaCl and mannitol were chosen as treatments for abiotic stress. The results demonstrated that salt and osmotic stress, induced by NaCl and mannitol treatments, could greatly enhance the positive rates of genetic transformation mediated by *A. rhizogenes* and the efficiency of CRISPR/Cas9-mediated genome editing in potato, although the regeneration of potato roots was inhibited to some extent (Tables 2 and 3). Specifically, the regeneration efficiency decreased by half at NaCl concentrations of 20 mM, and mannitol concentrations of 100 mM, respectively. Nevertheless, all regenerated roots contained mutations, which leads us to conclude that these are appropriate stress concentrations. Therefore, choosing the appropriate concentration of NaCl or mannitol stress can provide a balance between regeneration efficiency and genome editing efficiency, which is the trade-off between regeneration efficiency and genome editing efficiency under conditions of salt and osmotic stress. Genome-edited transformant plants can be obtained in large numbers by increasing the number of explants. In addition, this salt and osmotic stress-induced enhancement of the genetic transformation and genome editing efficiency could probably attribute to these abiotic stress-induced DNA damage due to intracellular accumulation of reactive oxygen species (*Chiera et al., 2008*; *Raja et al., 2017*), which accelerates non-homologous end joining (NHEJ). Future work is, however, needed to fully understand the effect of abiotic stress on genome editing efficiency.

CRISPR/Cas9-mediated genome editing systems have been reported to be able to generate null mutations in the T0 generation (*Zhang et al., 2014*). In this study, our analysis of the mutation types showed that chimeras accounted for the largest proportion, ranging from 62.50% to 100%, and that salt and osmotic stress could increase the rate of null mutations in potato (Fig. 4). Of all mutations, all insertion mutations were 1–2 bp, consistent with previous studies (*Pan et al., 2016*). Deletion mutations were predominantly small fragments, though we also found large fragment deletions of up to 213 bp (Fig. 5 and Table S1, M150-1).

Many studies have shown that CRISPR/Cas9-mediated genome editing technology can cause off-targeting (*Wu, Kriz & Sharp, 2014*; *Guo et al., 2023*). It has been demonstrated

that there is a seed region in the sgRNA proximal to the PAM, and the seed region requires at least 13 contiguous base pairs to pair with the target gene and complete cleavage (*Jinek et al., 2012*). None of the potential *StLike* off-target sites in the seed region can exactly match the selected sgRNA. As a result, no off-target effects were observed. We selected three off-target sites from the potential off-target sites for examination and did not find any off-targeting (Table 4), perhaps due to the high mismatch of the potential off-target sites (4 bp). However, it should be noted that we used Sanger sequencing, which could potentially lead to incomplete results, and some off-target effects may not have been detected.

## CONCLUSIONS

The present study clearly showed that appropriate salt and osmotic stress induced by NaCl and mannitol treatments could significantly enhance the positive rates of genetic transformation mediated by *A. rhizogenes*, as well as the efficiency of CRISPR/Cas9-mediated genome editing in potato, without any off-target effects. While these stresses inhibited the regeneration of potato roots to some extent, this trade-off is outweighed by the benefits. These findings provide a new and convenient approach to enhance the genome editing efficiency of CRISPR/Cas9-mediated genome editing systems in potato.

## ACKNOWLEDGEMENTS

We thank Dr. Dawei Li (Northwest Agriculture and Forestry University) for suggestions for modification of the vector of pKESE401.

### Funding

This work was financially supported by the National Natural Science Foundation of China (No. 32260459) and the open research program (No. YNPKF202202) of Yunnan Key Laboratory of Potato Biology, Yunnan Normal University. The funders had no role in study design, data collection and analysis, decision to publish, or preparation of the manuscript.

### Grant Disclosures

The following grant information was disclosed by the authors:
National Natural Science Foundation of China: No. 32260459.
Yunnan Normal University: No. YNPKF202202.

### Competing Interests

The authors declare that they have no competing interests.

### Author Contributions

- Mingwang Ye conceived and designed the experiments, performed the experiments, analyzed the data, prepared figures and/or tables, authored or reviewed drafts of the article, and approved the final draft.

- Mengfan Yao performed the experiments, prepared figures and/or tables, and approved the final draft.
- Canhui Li conceived and designed the experiments, authored or reviewed drafts of the article, and approved the final draft.
- Ming Gong conceived and designed the experiments, analyzed the data, authored or reviewed drafts of the article, and approved the final draft.

## Data Availability

The raw gel photo are available in the Supplemental File.

## Supplemental Information

Supplemental information for this article can be found online at http://dx.doi.org/10.7717/peerj.15771#supplemental-information.

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
