# Peer review of "Salt and osmotic stress can improve the editing efficiency of CRISPR/Cas9-mediated genome editing system in potato"

_PeerJ, doi:10.7717/peerj.15771_

## Round 0.1 · original submission · Major Revisions

Please provide a comprehensively revised version addressing the editorial comments and a detailed rebuttal letter.

Reviewer 1 ·

Basic reporting

Here, the authors evaluate the effect of salt and osmotic stress on the efficiency of CRISPR/Cas9-mediated genome editing in potatoes, demonstrating that this kind of stress could improve such efficiency. However, these treatments bring negative effects on the regeneration of potato roots. The work is well done, however, there are some points that concern me:

Minor comments:
Line 110: Do you mean “was digested”?
Line 123: Please italicize A. rhizogenes
Lines 239-244: You repeat several times this information. I suggest deleting this paragraph and moving it to the conclusions section.


Major comments:

Lines 39-50: This is interesting information; however, I think that does not contribute much to your work. Instead, I would prefer that you go deeper into the technical basis of CRISPR. Also, you didn’t discuss other works that have been done previously, besides yours. It is important to show the reader a broad and adequate context.

Line 103: Part of your work's justification is that you have variations between different potato germplasms. However, in this work, you only work with one of them. So, I think that you should have analyzed at least two different germplasms for your results to be more reliable.

Because you observed that the regeneration of potato root was inhibited, to what extent could your protocol work? Would it really be effective to use in practical terms?

In some points of the work, it seems to me that your results are preliminary. For example, what I told you about the germplasms. Also, you mention that is necessary to choose or standardize the suitable NaCl or mannitol concentration to reach a balance between root regeneration and genome editing efficiency of potato; however, I would have expected that in this work you could reach those results, at least under the conditions you are using.

Experimental design

No comments

Validity of the findings

No comments

Additional comments

No comments

Reviewer 2 ·

Basic reporting

The research document titled Salt and osmotic stress can improve the Editing Efficiency of CRISPR/Cas9-mediated genome editing system in Potato by Ye et al. aims to show the benefits of using salt or osmotic stress to enhance the efficiency of CRISPR/Cas9-mediated mutations in potato. The study positively presented results regarding the use of abiotic stresses during the transformation protocol. Mutations (INDELs and/or SNPs) in the StLike3 gene were obtained.
Altogether, this research article shows the benefit of using abiotic stresses to improve genome editing efficiency in potato. Specific suggestions would help the overall manuscript:

In all the manuscript check the following:

- When referring to “green fluorescence” throughout the manuscript, describing it as “gfp expression” would be better.
- The term “exogenous gene” is not appropriate for this study. The correct term would be “transgene” or “T-DNA”. Check this throughout the manuscript

Minor Reviews:

Abstract:
- Sentences in lines 23 and 32 are too long; it would be better if they were reworded and shortened.
- Line 25: stress-inducing; hyphen missing

Introduction:
- The word “fracture” does not fit the context which describes the DSB; it should be replaced with break or mutation depending on the sentence. (Example: lines 42 or 46)
- Line 49: the verb “be” is missing in: “can be knocked out”
- Line 49: The word “specificity” does not fit the grammar structure of the sentence, should “specific”
- Line 49-50: The structure of the last part of the sentence is not correct; it could be modified to “ in a targeted region for the rescue or replacement of gene function”
- Line 66: The word “and” is missing; “heat and cold”
- Lines 74-77: The sentence is too long, and the word “and” is repetitive. Each “and” could be a different sentence

Materials and Methods:
- Lines 115 and 119: Given that the reference is part of the sentence, it should be written without the brackets. Example “…backbone using the protocol described by Xing (2014)”. Check if this is the case for other sentences
- Line 137: the grammar structure for the sentence would be “…were amplified…”
- Line 144: “sites” shouldn’t be plural, given that there is only one target site in the study
- Line 146: “Spanning” is not the best fit. It could be reworded as “ The specific primers targeting potential OTs are shown in Table 1”

Results:
- Line 153: The gene “StLike3” should be in italics
- Line 156: the word “which” is not required in the sentence
- Lines 158 to 161: The sentence is too long; it can be split into two sentences.
- Line 188-191: the words “deletion mutation” could be just replaced with “deletions”. The same applies to “insertion mutation” – use Insertions. In the case of “replacement mutations” this would be better referred to as “SNPs”. Check this throughout the document.
- Line 195: The word “theses” should be “these abiotic”

Discussion:
- Lines 207 to 209: At the end of the sentence, maybe the word “respectively” is unnecessary. How the sentence is currently written implies that at 37⁰C, it was 5-fold, but at 22⁰C, it was 100-fold, which is the contrary of what the sentence states.
- Lines 217 to 219: The sentence structure and grammar are confusing, so it is unclear what they describe.
- Line 223: “More” is not needed

Figures and Tables:
- In Figure 1C: Instead of using lowercase letters, it could be better to label the figure with the treatments written in each panel. Also, a scale should be added to see the size in the picture of the root hair.
- In Table 1: Change “Detect…” to “Detection of…”
- In Table 2: Instead of “Transformant roots” used “Transformed roots”
- In Table 2: Instead of “Salt and osmotic stress” use “Treatments”
- In Table 3: Instead of “Stresses” use “Treatments”

Experimental design

Queries:

- What is the bacteria and plant selection used during transformation?
- In this study, the gene StLike3 gene is the target. Why was this gene picked? How many copies of this gene are present in the variety? What is the expected phenotype? Would this phenotype show even when not all copies are knocked out? or must it be a full knockout to show the phenotype?
- In the Materials and Methods, Line 147, why are those treatments (20 mM NaCl and 100 mM mannitol) used for the off-target assessment and not the others?
- For the results, It would be better if sequencing of the mutations is shown, especially from representative scenarios, for example, INDELs, SNPs, and chimeras
- In the Results section, when treatments are compared for efficiency and when types of mutation are compared. It would be better if a Student’s t-test is performed to see if there are significant differences among the treatments.
- In the Results section, the authors refer to “Positive rates”; how this is calculated is not explained anywhere. A positive rate is usually calculated as the part of the sample that shows an expected result divided by the total, is this the case? An explanation should be added in Materials and Methods
- In the Results, when referring to chimeras, does it mean there were more than two types of mutations when the DNA was sequenced? (assuming that the gene targeted has only two copies in the genome) Alternatively, if the sequencing data show a complete knockout but the phenotype does not, chimeric tissue is in the sample.
- The Discussion needs to be expanded. Add more explanation and more discussion regarding the two types of stress selected. Different kinds of abiotic stresses were used in other studies, and how they compare to the ones picked for this study. Are there studies with these specific stresses in different crops? Regarding the OTs: It can be added that some of the SNPs found in the potential OTs were located in the seed region, which decreases the possibility of Cas9 making an error with the corresponding references.
- Can more recent studies be used as part of the Discussion section?
- Paragraph (lines 225 to 233): Regarding null mutations (Absence of gene product), this needs clarification or rewording given that this can be seen when there are homozygous mutations, heterozygous mutations, and chimeric mutations.

Validity of the findings

no comment

---

## Round 0.2 · accepted · Accept

Thanks for addressing the minor revisions requested. Now your manuscript is accepted in PeerJ.

Reviewer 1 ·

Basic reporting

The authors considerably improved their manuscript, attended to the recommendations, and clarified some important points.

Experimental design

NA

Validity of the findings

NA

Additional comments

NA

Reviewer 2 ·

Basic reporting

Observations were taken into account, and changes were implemented in the manuscript.

Experimental design

Observations were taken into account, and changes were implemented in the manuscript.

Validity of the findings

N/A